# Air Quality in Brno City Parks

**Jiří Huzlík [1],\* , Jitka Hegrová [1] , Karel Effenberger [1] , Roman Ličbinský [1] and Martin Brtnický [2,3]**

1 Transport Research Centre, Lisenska 33a, 636 00 Brno, Czech Republic; jitka.hegrova@cdv.cz (J.H.); karel.effenberger@cdv.cz (K.E.); roman.licbinsky@cdv.cz (R.L.)
2 Department of Agrochemistry, Soil Science, Microbiology and Plant Nutrition, Faculty of AgriSciences, Mendel University in Brno, Zemedelska 1, 613 00 Brno, Czech Republic; martin.brtnicky@mendelu.cz
3 Institute of Chemistry and Technology of Environmental Protection, Brno University of Technology, Faculty of Chemistry, Purkynova 118, 621 00 Brno, Czech Republic
\* Correspondence: jiri.huzlik@cdv.cz

**Abstract:** Parks embody an important element of urban infrastructure and a basic type of public space that shapes the overall character of a city. They form a counterweight to built-up areas and public spaces with paved surfaces. In this context, parks compensate for the lack of natural, open landscapes in cities and thus have a fundamental impact on the quality of life of their inhabitants. For this reason, it is important to consider the quality of the environment in urban parks, air quality in particular. Concentrations of gaseous pollutants, namely, nitric oxide (NO), nitrogen dioxide ($NO_2$), and ozone ($O_3$), were measured in parks of Brno, the second-largest city in the Czech Republic. Relevant concentration values of $PM_{10}$ solids were determined continuously via the nephelometric method, followed by gravimetric method-based validation. The results obtained through the measurement of wind direction, wind speed, temperature, and relative humidity were used to identify potential sources of air pollution in parks. The "openair" and "openairmaps" packages from the OpenSource software R v. 3.6.2 were employed to analyze the effect of meteorological conditions on air pollution. Local polar concentration maps found use in localizing the most serious sources of air pollution within urban parks. The outcomes of the analyses show that the prevailing amount of the pollution determined at the measuring point most likely originates from the crossroads near the sampled localities. At the monitored spots, the maximum concentrations of pollutants are reached especially during the morning rush hour. The detailed time and spatial course of air pollution in the urban parks were indicated in the respective concentration maps capturing individual pollutants. Significantly increased concentrations of nitrogen oxides were established in a locality situated near a busy road (with the traffic intensity of 33,000 vehicles/d); this scenario generally applied to colder weather. The highest $PM_{10}$ concentrations were measured at the same location and at an average temperature that proved to be the lowest within the entire set of measurements. In the main city park, unlike other localities, higher concentrations of $PM_{10}$ were measured in warmer weather; such an effect was probably caused by the park being used to host barbecue parties.

**Keywords:** air pollution; urban parks; particulate matter; nitrogen oxides; ozone

## 1. Introduction

Urban green spaces, namely, city parks, are very often considered localities providing the best air quality in a city, and thus they become frequently targeted by citizens seeking relaxation and active recreation. However, there are very few studies supporting this generally accepted claim.

Xing et al. [1–3] noticed improved air quality in small urban parks within a distance from surrounding streets due to the dispersion of air pollutants within park areas. Importantly in this

context, trees can reduce wind speed and potentially trap pollutants. Most available studies point to a reduction of PM concentration levels inside city parks. Von Schneidemesser [4] stated that suitably distributed greenery can decrease the concentration of PMs by 20%, down to relative ambient average concentrations. Ou et al. [5] monitored $PM_{2.5}$ and $PM_{10}$ mass concentrations during the fall of 2018, identifying a significant drop in both $PM_{10}$ and $PM_{2.5}$ levels close to parks. A decrease of 23% in the total mass of $PM_{2.5}$ in a national park compared to an urban area is presented in paper [6]. Zhu et al. [7] analyzed the impact exerted by different types of plant communities on ambient $PM_{10}$ and $PM_{2.5}$ concentrations by using a spatial model. The results showed that differences in the levels of ambient PM concentrations among plant communities resulted from their composition and also other factors, including height (significantly lower ambient PM concentrations were recorded near small plants, namely, ones of less than 1 m), leaf area, or distance from the pollution source or edge of the park. Greenery increases the efficiency of reduction in ambient PM concentrations; however, this capability markedly depends on the season of the year. A significant decrease of $PM_{2.5}$ concentrations in La Carolina, a large city park in Ecuador, was described in [8]. Otosen et al. [9] measured differences in $PM_{1.0}$, $PM_{2.5}$, $PM_{10}$, $NO_2$, and CO concentrations in front of and behind vegetation barriers along roads (hedges during dormancy and the vegetative period). This type of greenery can mitigate the effects of air pollution generated by traffic, and, truly, a decrease in PM concentrations was measured. Contrariwise, no impact on the concentration of gasses was determined. In a relevant study by Abhijith and Kumar [10], the concentrations of $PM_{10}$, $PM_{2.5}$, $PM_{1.0}$, and black carbon were established in close vicinity of the three types of green infrastructure. The influence of separate hedges or shrubs, separate trees, and a mixture of trees and hedges/shrubs was assessed when located at different distances from a road, namely, at very close (<1 m from the road) and more remote (>2 m from the road) spots. The most prominent reductions were recorded in a mixture of trees and hedges under close distance conditions and in separate hedges positioned more remotely. An assessment of various PM fractions showed that separate hedges and a combination of trees and hedges decrease fine particle concentrations behind the green barrier. Relevant analyses then indicated a reduction of vehicle-related particles (i.e., those containing iron and its oxides, Ba, Cr, Mn) in the background of the green infrastructure, as compared to the front area. A similar paper on green infrastructure barriers, Mori et al. [11], characterized measurements of PMs sized between 0.2 μm and 100 μm. The authors described a reduction in PM particles at different distances from the road (measured by passive samplers), proposing that the actual results are influenced by different planting densities in two different green vegetation types of two heights.

Air pollution and human health, as well as green infrastructure and human health, are often studied together. Linking green infrastructure with air quality and human health is an aspect of interest for Kumar et al., who, in a corresponding review [12], concluded that although urban vegetation can bring health benefits, the knowledge of its wider applicability in efforts to reduce air pollution remains overly insufficient and must be further refined. Almedia et al. [13] discussed differences in pollutant concentrations ($PM_{10}$, $NO_2$, and $O_3$) between schools near roads in urban areas and schools adjacent to forests and roads in the same environment. The results correlate with respiratory problems exhibited by children within all areas of interest. The $PM_{10}$ and $NO_2$ concentrations proved to be higher at points closer to roads with intense traffic flows and lower at spots near parks with dense vegetation. Sheridan et al. [14] focused on $NO_x$ concentrations in the city of London, especially in parks and playgrounds, finding dangerously high levels of $NO_2$ at all places of interest (playgrounds, parks, and gardens), those open to the influx of the pollutant in particular. Lingberg et al. [15] described a reduction of air pollution in parks within the city of Gothenburg, Sweden; they emphasized the "park effect", namely, the assumption that parks embody a considerably cleaner local environment thanks to an interaction of two effects: dilution (the distance effect) and deposition. Trees and other vegetation can absorb and capture air pollutants, thus improving the air quality in cities. Due to a lack of local-scale information, the impact of urban parks and forest vegetation on the levels of nitrogen dioxide ($NO_2$) and ground-level ozone ($O_3$) were studied in Baltimore, USA. Yli-Pelkonen et al. [16]

concluded that $O_3$ concentrations were significantly lower in tree-covered habitats than in open ones. Conversely, $NO_2$ concentrations did not differ significantly between tree-covered and open habitats, meaning that it is again necessary to stress the choice and variability of greenery. Hewitt et al. [17] discussed several options of how to improve air quality by using different types of green infrastructure, introducing a novel conceptual framework as policy guidance; the authors' interpretation of the problem includes a flow chart to aid decision-making as regards the "green infrastructure to improve urban air quality".

Air pollution poses a major risk to human health, causing premature deaths and potentially reducing the quality of life. Quantifying the role of vegetation in curbing air pollution concentrations is an important step. Most current methods to calculate pollution cutback procedures are static and thus represent neither atmospheric transport of pollutants nor pollutants and meteorology interaction. The focus on urban parks as a tool to facilitate air purification and climate regulation embodies the basis of articles by Vieira et al. [18] and Mexia et al. [19]. These authors concluded that ecosystem service strongly depends on the vegetation type; thus, for example, air purification is more pronounced in mixed forest, and carbon reduction is influenced by tree density. Further, Jones et al. [20] developed a method to calculate health benefits directly from changes in pollutant (including $PM_{2.5}$, $NO_2$, $SO_2$, and $O_3$) concentrations, exploiting an atmospheric chemistry transport model.

In our paper, the concentrations of $PM_{10}$ solids were determined continuously, by utilizing the nephelometric method followed by gravimetric method-based validation. To identify potential sources of air pollution in parks, we evaluated the air quality within the local environment via correlation with measurements of wind direction, wind speed, temperature, and relative humidity. The "openair" and "openairmaps" packages from the OpenSource software R were employed to analyze the effects of wind on air pollution. Local polar concentration maps found application in locating the directions of wind coming from the most serious air pollution sources. Sampling and analyses were performed to confirm the assumption that the main sources of the pollution at the measuring point are most likely the roads and/or crossroads near the sampled localities.

Due to the information gap concerning air quality in city parks, the goal of our study was to obtain data on air pollution in urban parks and associated details relevant to the relationship between this pollution and meteorological parameters, prominently including temperature, wind speed, and wind direction; in this context, our efforts also involved comparing these data with pollution around the parks. Based on the findings, we then aimed to estimate the sources of air pollution in the monitored parks.

## 2. Method

### 2.1. Sampling

The sampling was carried out in three pre-selected city parks in Brno, the Czech Republic; two of the parks are located in areas with a high traffic impact (near main roads), while one is found in a low traffic load environment (a small park inside a courtyard). The main city park of Lužánky exhibits the largest surface area of all the monitored parks, and it is located near the city center, surrounded by roads with heavy traffic. Two air quality monitoring spots were positioned in the park: one place in the middle of the area, and the other on the edge of the park, near a playground and the traffic-laden roads. This park is frequently visited and used for sports and leisure activities, including picnics.

The Koliště park is adjacent to a road with heavy traffic (33,000 cars/d). It occupies a large walking-friendly area, and there is a very popular restaurant in the middle of the park. However, due to the traffic-laden road, the location is not a popular target for sports, children's activities, or picnics. The air quality was measured near a junction of two main roads.

Tyršův sad is a very small park in the city center, situated inside a courtyard. This park is mostly used only for short walks, especially with dogs. The air quality measurement was performed in the middle of the area.

The devices were installed together at the place of interest.

## 2.2. Instrumentation

The NO, $NO_2$, $O_3$, and $PM_{10}$ concentrations were determined by using two Airpointer units (Recordum Messtechnik GmbH, Austria). These devices measure pollutant concentrations via separate modules utilizing type-approved reference methods ($NO_2/NO_X$, $O_3$) classified as relevant by the EU, WHO, US-EPA, and other competent responsible organizations worldwide.

The measurement principle to define the levels of $NO_2/NO_x$ is chemiluminescence (EN14211). The Airpointer $NO_X$ module was equipped with a delay loop to measure NO and $NO_2$ from the same sample. An external calibration gas with a concentration of 425 ppb NO in $N_2$ (SIAD, Italy) was employed to periodically check the span point.

The $O_3$ measurement principally exploits UV absorption (EN 14625); for the given purpose, an internal ozone generator to allow regular span point checking was applied.

The parameters are calibrated annually by the Slovak Hydrometeorological Institute.

The Airpointer $PM_{10}$ module utilizes nephelometry for measuring solid particles' concentrations. Gravimetric measurements of $PM_{10}$ concentrations executed within 24 h intervals were carried out to calibrate the nephelometric method. Sequential samplers SVEN LECKEL SEQ 47/50-CD (Sven Leckel Ingenieurbüro GmbH, Germany) were employed for the calibration. The particles were collected on cellulose nitrate filters with the porosity of 1.2 µm (Merck, Germany) and weighed on a Mettler Toledo MX/A microbalance.

The meteorological parameters (air temperature, relative humidity, air pressure, wind speed, and wind direction) were measured by using a compact meteorological station integrated with the Airpointer. These parameters are regularly calibrated by the Czech Metrology Institute.

The data from the Airpointer were downloaded as CSV files and saved in the form of Microsoft Excel files (XLSX). The concentrations measured in ppb were converted to concentrations in µg m$^{-3}$. The medians, upper and lower quartiles, and other percentiles for the monitored pollutants, temperature, relative humidity, and wind speed were calculated in MS Excel. The results were then processed by the Origin program (OriginLab, USA) to yield graphs. The dependencies of and relationships between the pollutant concentrations on the wind speed and direction were processed via the "openair" and "openairmaps" packages of OpenSource program R [21,22]. The package "openairmaps" supports "openair" for plotting on various maps. The maps include those available via the "ggmap" package, e.g., Google Maps, and leaflet ones to facilitate plotting bivariate polar plots. Our research utilized the "Esri.WorldImagery" map source and the "Non-parametric Wind Regression" (NWR) technique to display the concentration maps as bivariate polar plots.

## 2.3. Measurement Conditions and Positioning of Instruments

The concentrations of $PM_{10}$ and also those of the gaseous pollutants NO, $NO_2$, and $O_3$ in three parks within the city of Brno, the Czech Republic, were measured in one-minute intervals. The same scenario was applied to the meteorological conditions, namely, air temperature (T), relative humidity (RH), air pressure (p), wind speed (WS), and wind direction (WD). The $NO_2$ and $PM_{10}$ measurements at automated air pollution monitoring stations operated by the Czech Hydrometeorological Institute were employed for comparing the measurement results with those acquired at a heavy traffic locality (Údolní, the Hot Spot) and background localities (Arboretum—the natural city background station, and Dětská nemocnice—the commercial city background station). Tables 1 and 2 show the geographic coordinates of the localities and display the time intervals of the measurement.

**Table 1.** The geographic coordinates of the measured localities.

| Sampling Locality | Latitude ° N | Longitude ° E |
|---|---|---|
| Tyršův sad | 49.2027128 | 16.6023589 |
| Lužánky SS [1] | 49.2083389 | 16.6077778 |
| Lužánky SVC [2] | 49.2065792 | 16.6069417 |
| Koliště | 49.1966892 | 16.6145658 |
| Koliště road [3] | 49.1970100 | 16.6147853 |
| Úvoz Hot Spot | 49.1980897 | 16.5936431 |
| Arboretum | 49.2160872 | 16.6138364 |
| Dětská nemocnice | 49.2027244 | 16.6162872 |

[1] Site Svojsík's Cabin. [2] Site Leisure Centre. [3] Site alongside an adjacent roadway.

**Table 2.** The measurement times related to the localities.

| Campaign Start | Campaign End | Lužánky SS [1] | Lužánky SVC [2] | Tyršův Sad | Koliště | Úvoz | Dětská n. | Arboretum | Koliště Road [3] |
|---|---|---|---|---|---|---|---|---|---|
| 12.9.2018 12:00 | 26.9.2018 11:59 | | x | | | | | | |
| 18.1.2019 7:00 | 1.2.2019 6:59 | | | | x | x | x | x | |
| 8.2.2019 7:00 | 22.2.2019 6:59 | | | x | | x | x | x | |
| 6.3.2019 7:00 | 20.3.2019 6:59 | x | x | | | x | x | x | |
| 7.6.2019 7:00 | 21.6.2019 6:59 | | | | x | x | x | x | |
| 2.8.2019 7:00 | 16.8.2019 6:59 | | | x | | | | | |
| 22.8.2019 7:00 | 5.9.2019 6:59 | x | x | | | | | | |
| 8.11.2019 0:00 | 25.11.2019 23:59 | | | | | | | | x |

[1] Site Svojsík's Cabin. [2] Site Leisure Centre. [3] Site alongside an adjacent roadway.

Figures 1–3 show the positions of the measurement devices at the sampling sites. The devices were secured against theft with chains and connected to a power supply with a cable. The progress of the measurement was checked via an Internet connection through a SIM card.

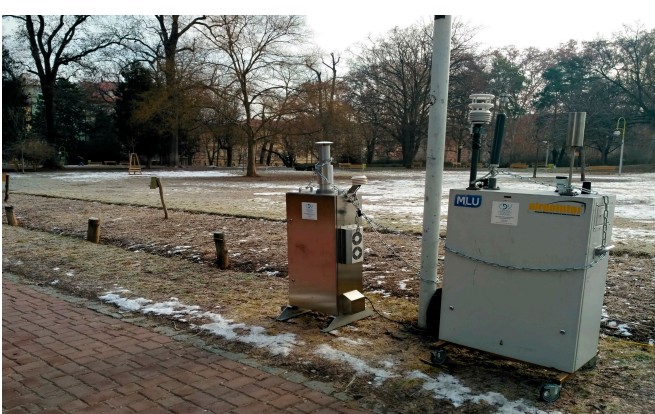

**Figure 1.** The devices at Tyršův sad.

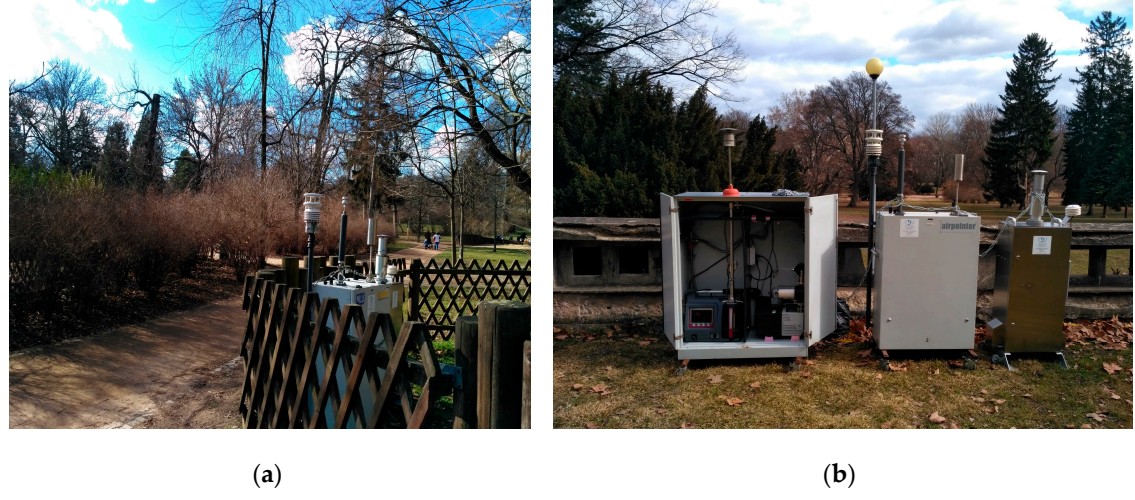

(**a**)　　　　　　　　　　　　　　　　　　　　(**b**)

**Figure 2.** The devices at Lužánky: (**a**) Svojsík's Cabin; (**b**) Leisure Centre.

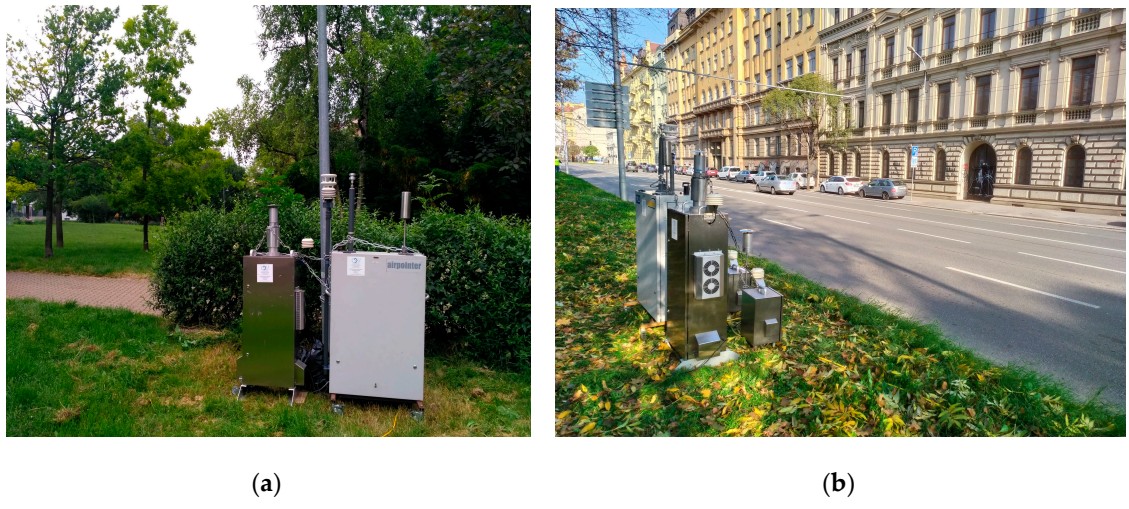

(**a**)　　　　　　　　　　　　　　　　　　　　(**b**)

**Figure 3.** The devices at Koliště: (**a**) Inside the park; (**b**) at the adjacent roadway.

Figure 4 shows the location of the sampling sites on a map of Brno.

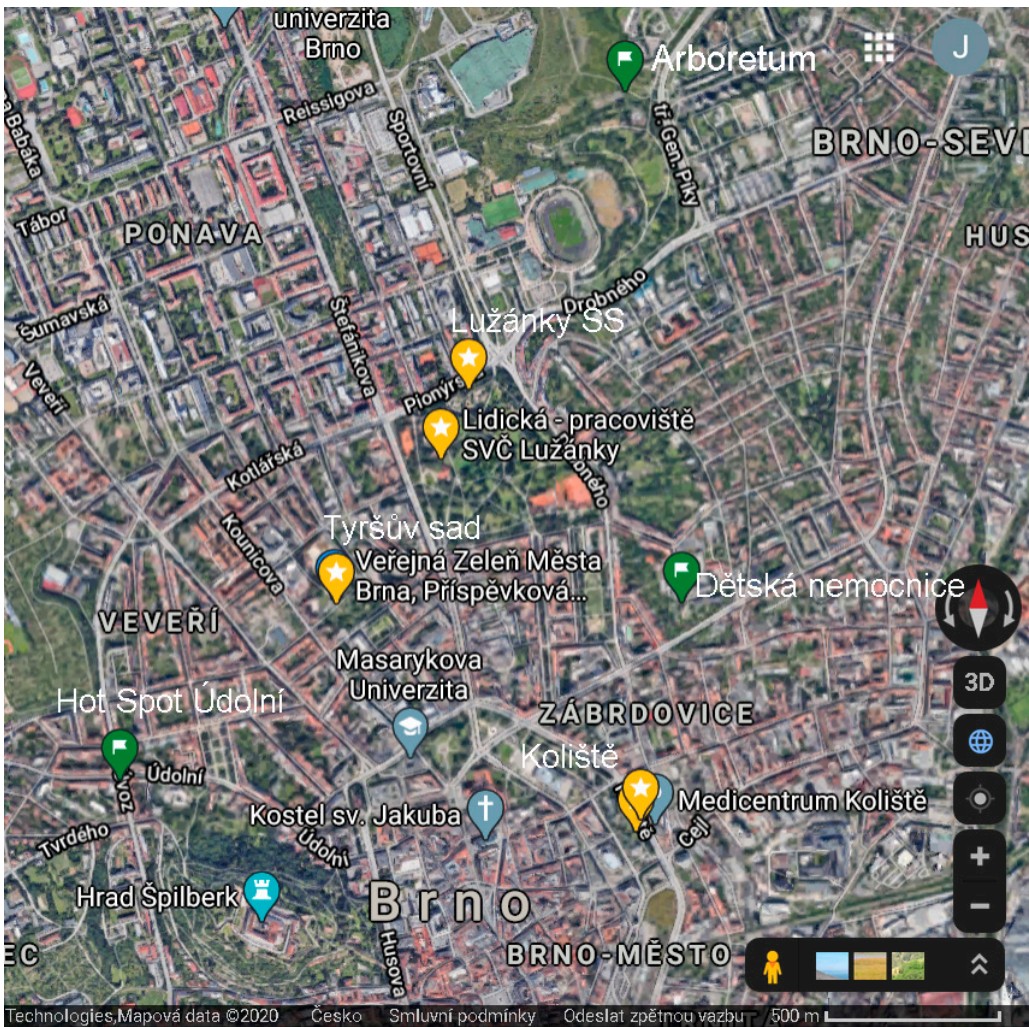

**Figure 4.** The sampling (yellow marks) and reference (green marks) localities.

### 2.4. PM₁₀ Calibration

As nephelometric measurements are performed in one-minute intervals, the conversion factor was calculated for each 24 h measurement interval according to the formula

$$f = \frac{PM_{10}^{grav}}{\overline{PM_{10}^{neph}}} \tag{1}$$

where

$PM_{10}^{grav}$ is the gravimetric PM₁₀ concentration over 24 h (μg/m³), and

$\overline{PM_{10}^{neph}}$ is the average nephelometric PM₁₀ concentration over 24 h (μg/m³)

The calculated emission factor is discontinuous, and was thus smoothed by the function

$$Factor = f_i + \frac{f_{i+1} - f_i}{2} \times \left( tgh\left[ p \times \left( t - t_{day} \times floor\left( \frac{t + \frac{t_{day}}{2}}{t_{day}} \right) \right) \right] + 1 \right), \tag{2}$$

where

*Factor* is the smoothed conversion factor in time *t*

$f_i$ is the conversion factor for the $i$th day

$f_{i+1}$ is the conversion factor for the $(i + 1)$th day

$p$ is the smoothing parameter ($p = 0.004$)

$t$ is the time from the start of the measurement (minutes)

$t_{day}$ is the length of the day (minutes)

*floor*() is the rounding down function

*tgh*[] is the hyperbolic tangent function

The $PM_{10}$ concentration was calculated for every minute by the function

$$PM_{10} = Factor \times PM_{10}^{neph}. \tag{3}$$

An example of the factors' calculation for the site Tyršův sad is shown in Figure 5.

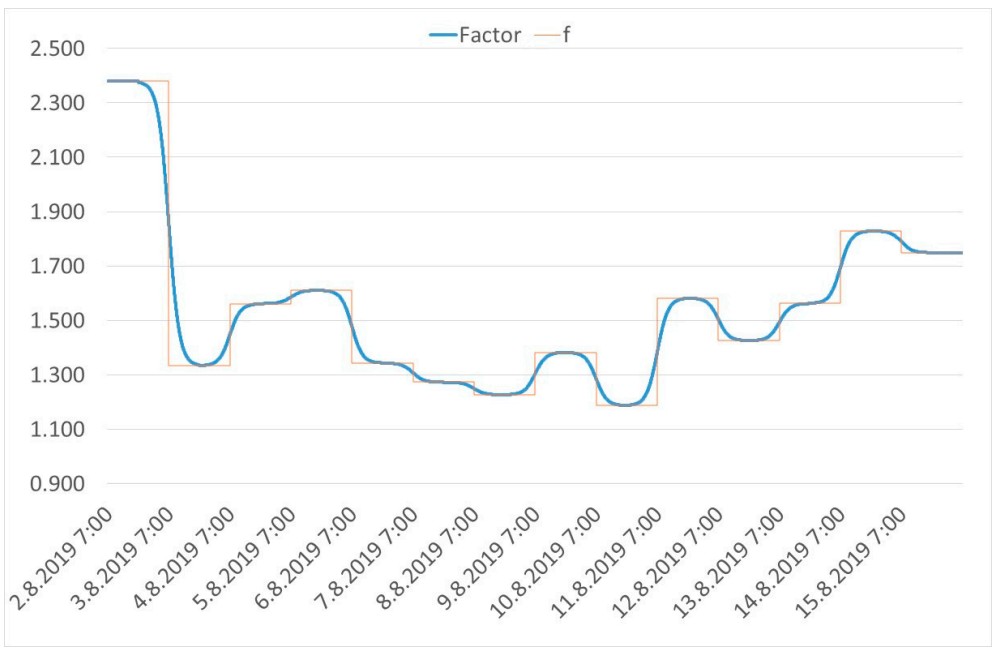

**Figure 5.** A comparison of the factors for the locality Tyršův sad from 2.8.2019 7:00:00 to 16.8.2019 6:59:00.

## 3. Results and Discussion

Each measurement at a park is represented by a dataset with 20,160 observations, and the measurement at the Koliště road locality is represented by a dataset with 25,920 observations. Therefore, the results were summarized as percentiles and mean values to be calculated in MS Excel. Table 3 shows the intervals in which 90% of the measured values are considered for each parameter.

**Table 3.** The measurement results: the 0.05 and 0.95 percentiles of the measured parameters.

| Site | Start | Percentile | NO µg/m³ | NO₂ µg/m³ | NOₓ µg/m³ | PM₁₀ µg/m³ | O₃ µg/m³ | T °C | RH % | WS m/s |
|------|-------|-----------|----------|-----------|-----------|------------|----------|------|------|--------|
| Tyršův sad | 8.2.2019 7:00 | 0.05 | 0.92 | 8.90 | 10.89 | 4.05 | 3.42 | −1.16 | 55.50 | 0.00 |
| | | 0.95 | 133.54 | 76.87 | 286.17 | 75.44 | 116.78 | 9.56 | 92.33 | 0.90 |
| Tyršův sad | 2.8.2019 7:00 | 0.05 | 0.50 | 2.77 | 3.99 | 6.05 | 17.43 | 13.96 | 40.54 | 0.00 |
| | | 0.95 | 4.80 | 18.66 | 25.09 | 17.05 | 102.21 | 28.22 | 94.27 | 0.80 |
| Lužánky SVC | 12.9.2018 12:00 | 0.05 | 1.23 | 4.10 | 6.67 | 3.10 | 1.01 | 5.52 | 44.48 | 0.00 |
| | | 0.95 | 40.74 | 54.41 | 114.90 | 29.45 | 100.47 | 25.91 | 100.00 | 1.52 |

**Table 3.** *Cont.*

| Site | Start | Percentile | NO μg/m³ | NO₂ μg/m³ | NOₓ μg/m³ | PM₁₀ μg/m³ | O₃ μg/m³ | T °C | RH % | WS m/s |
|---|---|---|---|---|---|---|---|---|---|---|
| Lužánky SVC | 6.3.2019 7:00 | 0.05 | 2.13 | 3.43 | 6.95 | 7.65 | 5.48 | 0.53 | 42.93 | 0.00 |
| | | 0.95 | 21.71 | 50.43 | 86.07 | 33.21 | 112.67 | 12.90 | 100.00 | 1.61 |
| Lužánky SVC | 22.8.2019 7:00 | 0.05 | 0.74 | 4.42 | 6.03 | 11.28 | 5.49 | 11.41 | 41.59 | 0.00 |
| | | 0.95 | 18.23 | 39.68 | 67.52 | 53.80 | 116.80 | 29.28 | 93.92 | 1.14 |
| Lužánky SS | 6.3.2019 7:00 | 0.05 | 1.18 | 4.67 | 6.84 | 4.64 | 2.75 | −0.15 | 43.38 | 0.00 |
| | | 0.95 | 38.42 | 55.78 | 115.50 | 35.88 | 130.53 | 12.90 | 93.55 | 1.15 |
| Lužánky SS | 22.8.2019 7:00 | 0.05 | 1.55 | 6.52 | 9.56 | 16.27 | 1.77 | 11.76 | 41.54 | 0.00 |
| | | 0.95 | 29.27 | 40.73 | 86.31 | 78.20 | 112.04 | 29.62 | 100.00 | 0.82 |
| Koliště | 18.1.2019 7:00 | 0.05 | 1.22 | 17.67 | 20.26 | 24.19 | 4.46 | −8.68 | 58.53 | 0.00 |
| | | 0.95 | 136.72 | 87.80 | 291.13 | 134.13 | 81.96 | 1.58 | 92.78 | 1.38 |
| Koliště | 7.6.2019 7:00 | 0.05 | 0.92 | 7.06 | 9.85 | 6.06 | 19.39 | 16.37 | 43.15 | 0.00 |
| | | 0.95 | 8.12 | 36.20 | 48.46 | 55.14 | 122.94 | 30.44 | 99.10 | 1.25 |
| Koliště road | 8.11.2019 0:00 | 0.05 | 1.85 | 11.34 | 15.00 | 21.61 | 2.96 | 2.34 | 73.24 | 0.00 |
| | | 0.95 | 153.88 | 61.53 | 288.60 | 93.26 | 34.30 | 12.36 | 94.73 | 1.74 |

The results of the measurements at the automatic air pollution monitoring stations were used to compare the air pollution concentrations in the parks and their vicinity. The hourly averages of the NO₂ and PM₁₀ concentrations were compared, as the data are measured in hourly intervals. The results are shown in Table 4.

**Table 4.** The results of the measurement at the Automated Air Pollution Monitoring Stations—the 0.05 and 0.95 percentiles of the measured concentrations.

| Automated Air Pollution Monitoring Station | Start | Percentile | NO₂ μg/m³ | PM₁₀ μg/m³ |
|---|---|---|---|---|
| Úvoz | 18.1.2019 7:00 | 0.05 | 23.01 | 24.75 |
| | | 0.95 | 86.46 | 119.25 |
| Arboretum | 18.1.2019 7:00 | 0.05 | 13.60 | 13.55 |
| | | 0.95 | 60.65 | 101.23 |
| Dětská nemocnice | 18.1.2019 7:00 | 0.05 | 13.60 | 13.75 |
| | | 0.95 | 79.80 | 108.00 |
| Úvoz | 8.2.2019 7:00 | 0.05 | 19.17 | 8.75 |
| | | 0.95 | 82.08 | 91.75 |
| Arboretum | 8.2.2019 7:00 | 0.05 | 12.95 | 8.20 |
| | | 0.95 | 53.23 | 76.00 |
| Dětská nemocnice | 8.2.2019 7:00 | 0.05 | 10.90 | 8.00 |
| | | 0.95 | 80.90 | 76.50 |
| Úvoz | 6.3.2019 7:00 | 0.05 | 8.43 | 2.00 |
| | | 0.95 | 67.29 | 67.25 |
| Arboretum | 6.3.2019 7:00 | 0.05 | 6.04 | 2.50 |
| | | 0.95 | 40.88 | 27.65 |
| Dětská nemocnice | 6.3.2019 7:00 | 0.05 | 3.39 | 3.00 |
| | | 0.95 | 54.72 | 28.00 |
| Úvoz | 7.6.2019 7:00 | 0.05 | 9.84 | 10.00 |
| | | 0.95 | 75.16 | 44.00 |
| Arboretum | 7.6.2019 7:00 | 0.05 | 4.80 | 8.58 |
| | | 0.95 | 26.67 | 36.90 |
| Dětská nemocnice | 7.6.2019 7:00 | 0.05 | 3.40 | 8.00 |
| | | 0.95 | 46.70 | 40.25 |

Figure 6 compares the individual mean values (medians) of the measured pollutant concentrations and meteorological parameters. The dispersions of these values are represented through the upper and lower quartiles, an interpretation that is more plausible than that rendered via the mean and standard deviations because the data have an asymmetric statistical distribution. This is also clearly seen in Figure 6: the vertical lines, whose length represents the size of the first and the third quartiles, are not identically long.

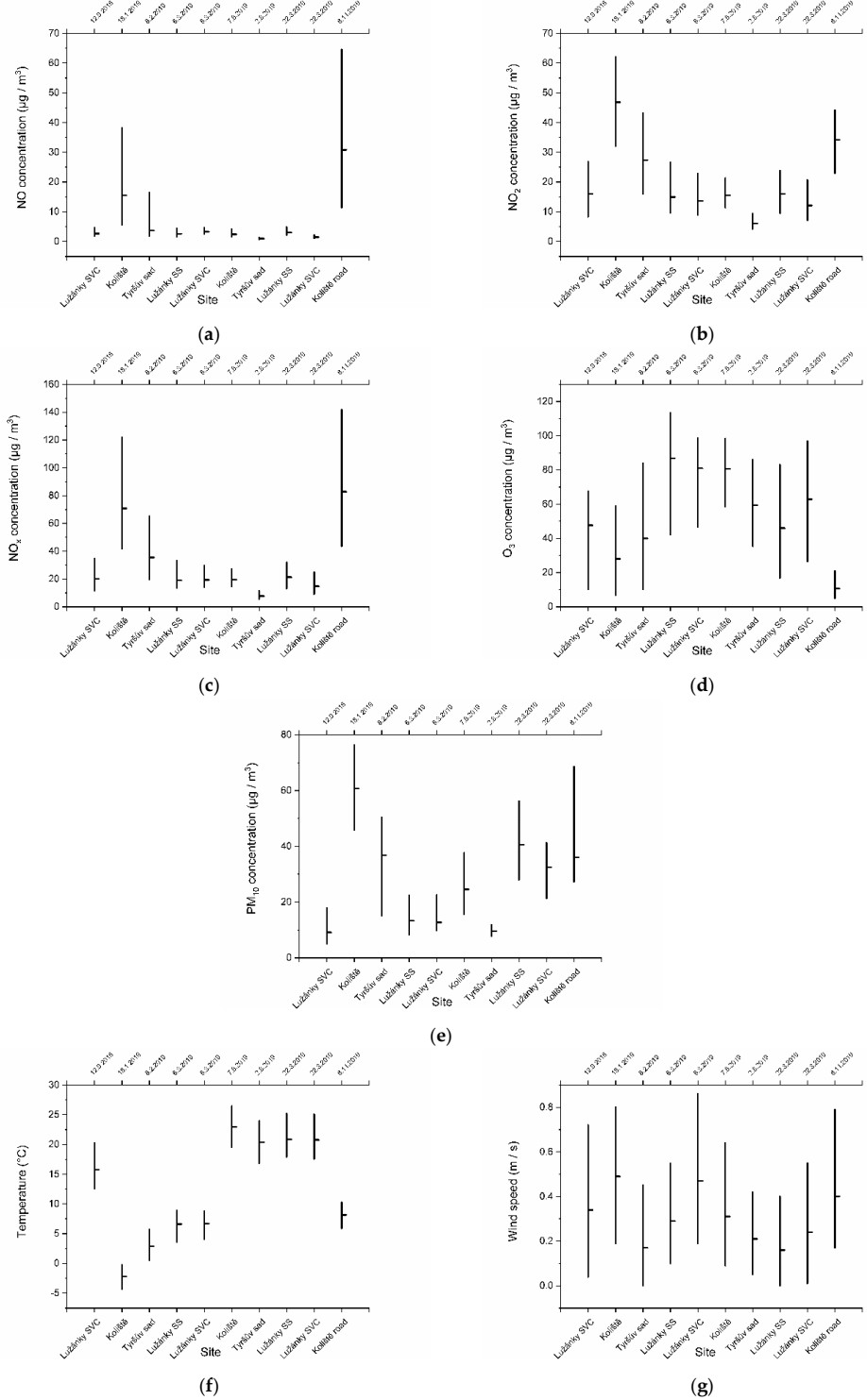

**Figure 6.** The NO (**a**), NO$_2$ (**b**), NO$_x$ (**c**), O$_3$ (**d**), and PM$_{10}$ (**e**) concentrations and temperature (**f**) plus wind speed (**g**) measured in the parks: the medians and quartiles.

The highest NO concentrations were measured in the immediate vicinity of the road adjacent to the Koliště park and then directly in the park; in both cases, the measurement was performed during a cold season (January, November). Similarly, the highest $NO_2$ concentrations were determined next to the road adjacent to the Koliště park and directly in the park (but also in Tyršův sad); in all of the cases, the measurement was carried out during a cold season (January, November, February). The highest total concentrations of nitrogen oxides ($NO_x$) were acquired, as in the NO, in the immediate vicinity of the road adjacent to the Koliště park and directly in the park, during a cold season (January, November). The highest $O_3$ concentrations were measured in springtime, the lowest one in winter. The solid particles detected at Lužánky SVC and Lužánky SS exhibited a higher concentration in August than in the colder months (March, September), which is not a normal effect. This deviation arises from the fact that, in these localities, people often gather for barbecue parties and use the parks' public cooking facilities during the summer months, whereas the other parks are not frequented for this purpose.

Figure 6f,g shows also the differences between the speeds and variations between the temperatures at the sampling sites, respectively. The March, February, and January temperatures reached significantly lower than the September, August, and June ones.

Figure 7a compares graphically the $NO_2$ air pollution in the parks, with the pollution measured at the reference stations, while Figure 7b displays, in the same manner, the air pollution caused by $PM_{10}$. The individual measurement campaigns are separated by the red lines. As can be seen, the air pollution in the parks, with the exception of the Koliště park for $PM_{10}$, was lower than that at the traffic locality, and the pollution at the background localities approached the value. The exception concerning the Koliště park was probably due to the fact that this area is relatively narrow compared to Lužánky; thus, in wintertime, when the vegetation is leafless, it provides less from the dust generated on the nearby busy roads. Moreover, it is obvious from the representation that the parks ensure better air protection from nitrogen oxides than against dust.

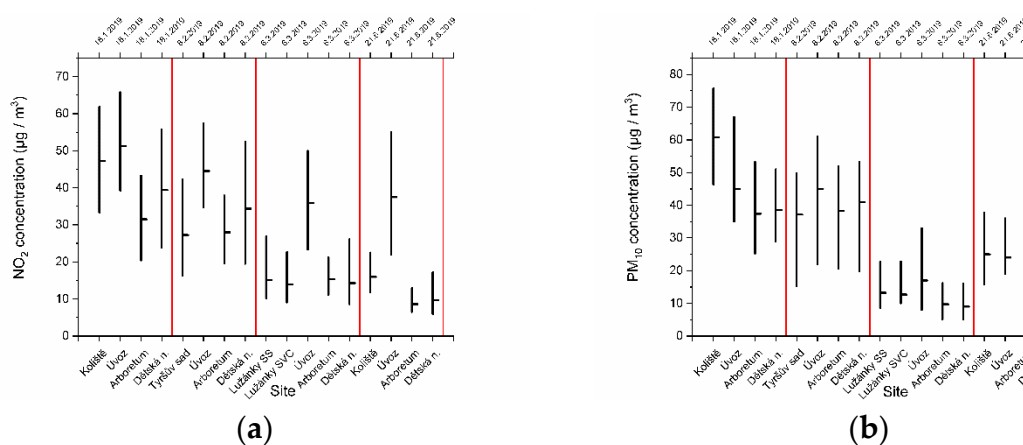

**Figure 7.** The $NO_2$ (**a**) and $PM_{10}$ (**b**) concentrations in the parks and at the automated air pollution monitoring sites.

The average concentrations of the measured pollutants were compared with the legal limits [23,24], Table 5. The excess values are marked in pink.

**Table 5.** The average concentrations from the measured localities as compared with the legal limits.

| Site | Measurement Start | NO$_2$ µg·m$^{-3}$ | NO$_x$ µg·m$^{-3}$ | PM$_{10}$ µg·m$^{-3}$ | O$_3$ mg·m$^{-3}$ | NO$_2$ | PM$_{10}$ Count [4] | O$_3$ |
|---|---|---|---|---|---|---|---|---|
| Lužánky SVC | 12.9.2018 12:00 | 20.52 | 32.74 | 12.01 | 44.95 | 0 | 0 | 0 |
| Lužánky SVC | 6.3.2019 7:00 | 18.41 | 28.96 | 16.60 | 71.44 | 0 | 0 | 0 |
| Lužánky SVC | 22.8.2019 7:00 | 16.01 | 22.22 | 33.57 | 61.48 | 0 | 2 | 3 |
| Lužánky SS | 6.3.2019 7:00 | 20.46 | 33.65 | 16.16 | 77.10 | 0 | 0 | 20 |
| Lužánky SS | 22.8.2019 7:00 | 18.53 | 30.31 | 43.46 | 50.46 | 0 | 4 | 0 |
| Koliště | 18.1.2019 7:00 | 48.89 | 100.97 | 66.60 | 34.47 | 0 | 11 | 0 |
| Koliště | 7.6.2019 7:00 | 17.66 | 22.94 | 27.16 | 77.51 | 0 | 0 | 14 |
| Koliště road | 8.11.2019 0:00 | 34.67 | 108.83 | 47.72 | 14.13 | 0 | 4 | 0 |
| Tyršův sad | 8.2.2019 7:00 | 32.95 | 67.61 | 36.20 | 48.78 | 0 | 4 | 0 |
| Tyršův sad | 2.8.2019 7:00 | 7.86 | 10.18 | 10.20 | 59.95 | 0 | 0 | 0 |
| **Legal limits** | | | | | | | | |
| Annual average limit | | 40 [1] | 30 [2] | 40 [1] | | | | |
| Day average limit [1] | | | | | | | 50 | |
| Day average count [1] | | | | | | | 35 | |
| Hourly average limit [2] | | | | | | 200 | | |
| Hourly average count [2] | | | | | | 18 | | |
| Max 8 h average limit [3] | | | | | | | | 120 |
| Max 8 h average count [3] | | | | | | | | 25 |

[1] Human health protection. [2] Ecosystems and vegetation protection. [3] Limit for tropospheric ozone. [4] Count of legal limit excess instances.

It is possible to claim that in most localities the NO$_x$ limit for ecosystems and vegetation protection was exceeded, except for Lužánky SVC in March and August 2019, Tyršův sad in August 2019, and Koliště in June 2019. The NO$_2$ concentration reached beyond the human health protection limit only in January 2019, when the lowest average temperature of all measurement campaigns was recorded. The PM$_{10}$ concentrations exceeded the same limit only at Koliště in January 2019, Koliště road in November 2019, and Lužánky SS in August 2019.

The analysis of the relationship between the individual pollutants' concentrations, wind speed, and wind direction was utilized to identify the places from which the highest pollutant concentrations reached the sampling site. The concentration scale of the measured pollutants is shown in Figure 12.

The color scale shown in Figure 8 expresses concentrations depending on wind direction (angle coordinate) and wind speed (radius coordinate). Figure 9, Figure 10, Figure 11, Figure 12, Figure 13, Figure 14, Figure 15, Figure 16, Figure 17, Figure 18 introduce the concentration polar maps of the measured pollutants at all of the localities.

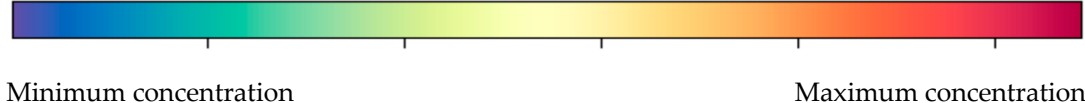

Minimum concentration                                                                 Maximum concentration

**Figure 8.** The concentration scale for the "openmaps" graphs.

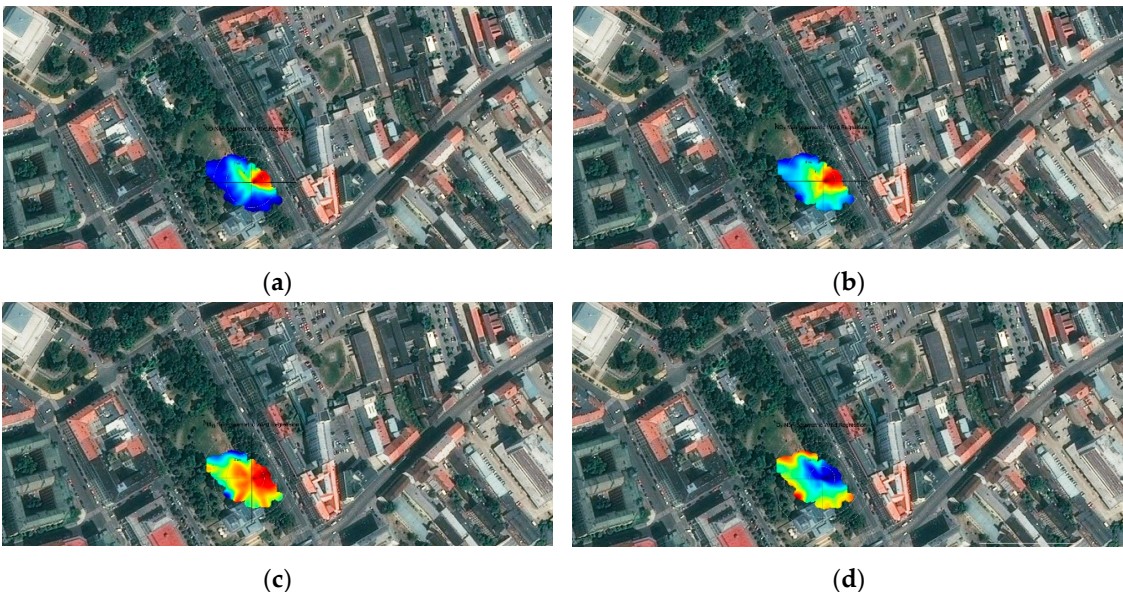

**Figure 9.** The NO (**a**), NO$_2$ (**b**), PM$_{10}$ (**c**), and O$_3$ (**d**) concentration relationships to the wind speed and direction at Koliště; sampling started on 18.1.2019 7:00.

Figure 9a,b shows that the highest concentrations of nitrogen oxides were measured with an east wind blowing from the adjacent road. Under the east to northwest wind direction, the lowest ozone concentrations were measured (Figure 9d). The lowest PM$_{10}$ concentration was obtained in north and south winds, meaning that transport embodies the most likely source of the nitrogen oxides; there is a larger amount of PM$_{10}$ sources; and, probably, the activities pursued within the area contribute to the dust circulation in the park (Figure 9c).

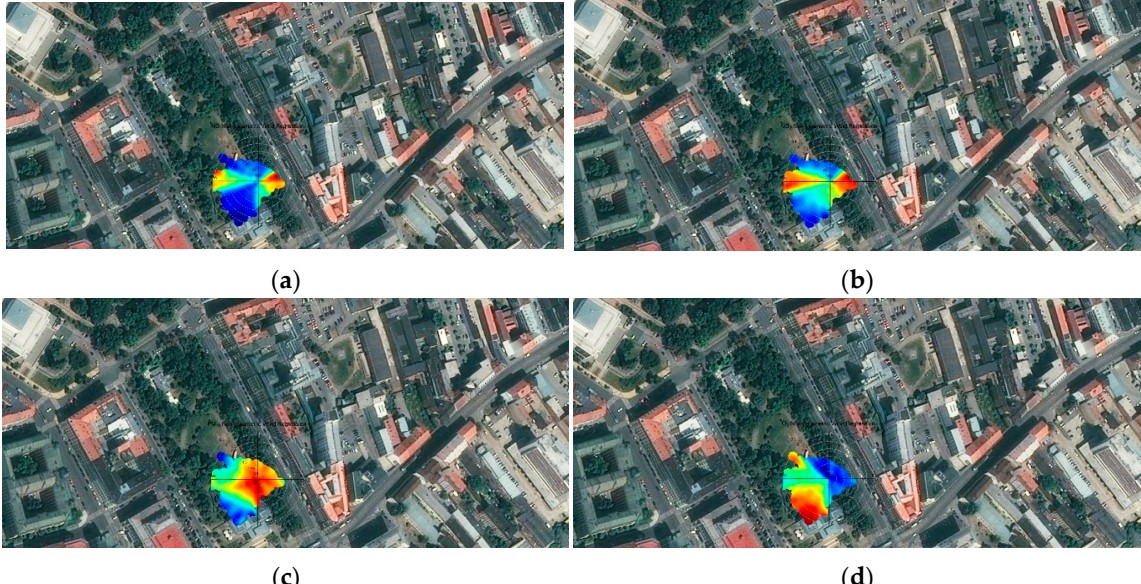

**Figure 10.** The NO (**a**), NO$_2$ (**b**), PM$_{10}$ (**c**), and O$_3$ (**d**) concentration relationships to the wind speed and direction at Koliště; sampling started on 7.6.2019 7:00.

Figure 10a,b shows that the highest concentrations of nitrogen oxides were measured with east and west winds blowing from the adjacent road and the opposite side. The impact of traffic on the road west of the park, which had not manifested itself in January, probably shows here. From the

south through the east to the northwest, the lowest ozone concentrations were measured (Figure 10d). The highest PM$_{10}$ concentrations were acquired in calm weather.

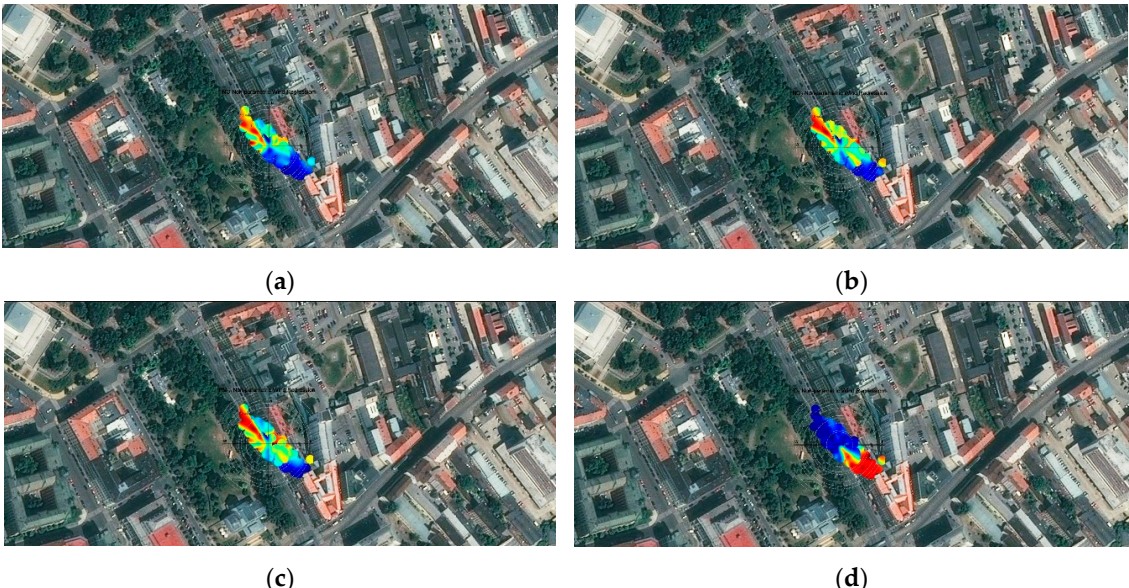

**Figure 11.** The NO (**a**), NO$_2$ (**b**), PM$_{10}$ (**c**), and O$_3$ (**d**) concentration relationships to the wind speed and direction at Koliště road; sampling started on 8.11.2019 0:00.

Figure 11a–c shows that the highest concentrations of nitrogen oxides and PM$_{10}$ were measured with northwest wind blowing in the direction of the vehicles traveling towards the Airpointer along the near lane of the road. At the same wind direction, we measured the lowest concentrations of O$_3$ (Figure 11d), meaning that both the oxides of nitrogen and the PM$_{10}$ had most likely originated from traffic in this case.

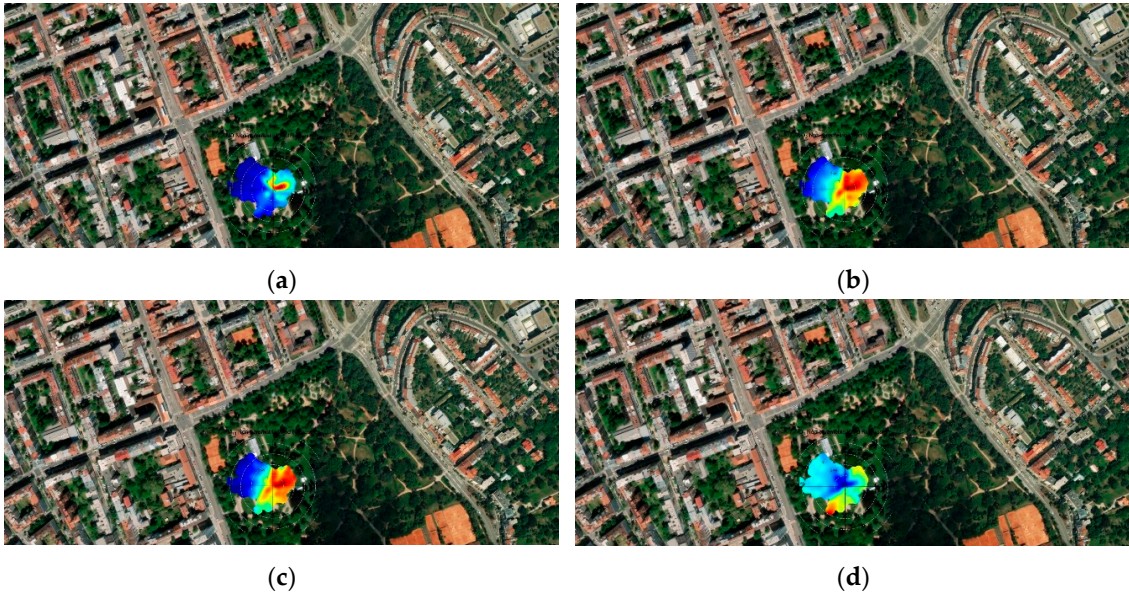

**Figure 12.** The NO (**a**), NO$_2$ (**b**), PM$_{10}$ (**c**), and O$_3$ (**d**) concentration relationships to the wind speed and direction at Lužánky SVC; sampling started on 12.9.2018 12:00.

Figure 12a indicates that the highest NO concentrations were measured with a east wind. The highest NO$_2$ concentrations were determined in eastern wind directions, namely, from the south to the north, similarly to PM$_{10}$ (Figure 12b,c). At low wind speeds, we acquired the lowest O$_3$

concentrations of the (Figure 12d), meaning that both the NO$_2$ and the PM$_{10}$ had probably been generated by similar sources. The NO had most likely originated from the traffic on the road east of the park.

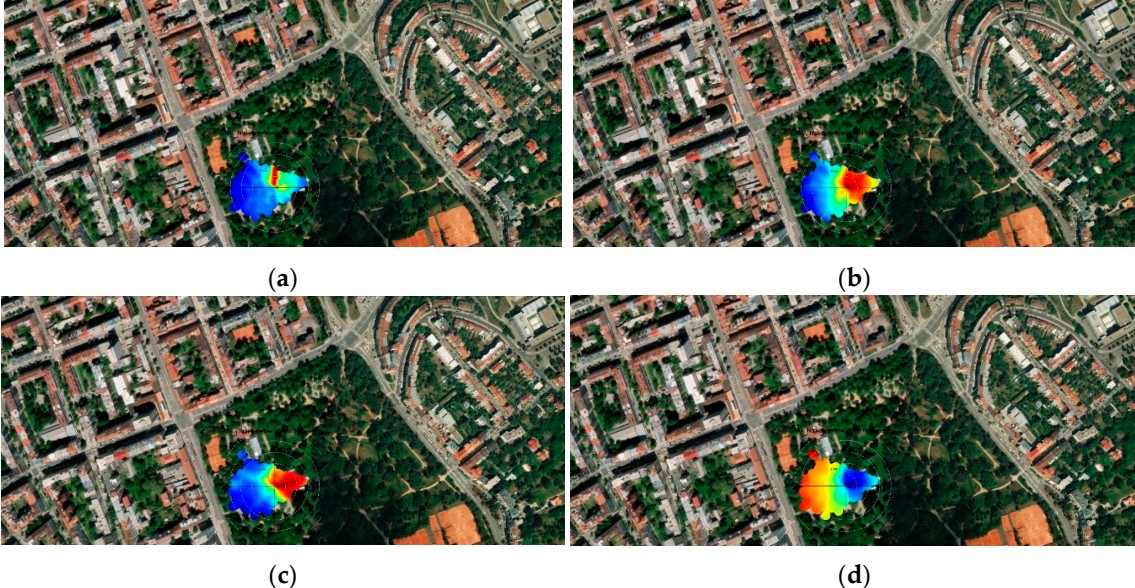

**Figure 13.** The NO (**a**), NO$_2$ (**b**), PM$_{10}$ (**c**), and O$_3$ (**d**) concentration relationships to the wind speed and direction at Lužánky SVC; sampling started on 6.3.2019 7:00.

Figure 13a indicates that the highest NO concentrations were measured with a north wind, similarly to the situation in Figure 14. The highest NO$_2$ concentrations were acquired under eastern wind directions, namely, from the south to the north, similarly to PM$_{10}$ (Figure 13b,c). This scenario resembles that represented in Figure 16. In eastern wind directions, we measured the lowest O$_3$ concentrations (Figure 13d). The NO had probably originated from the traffic on the road north of the park.

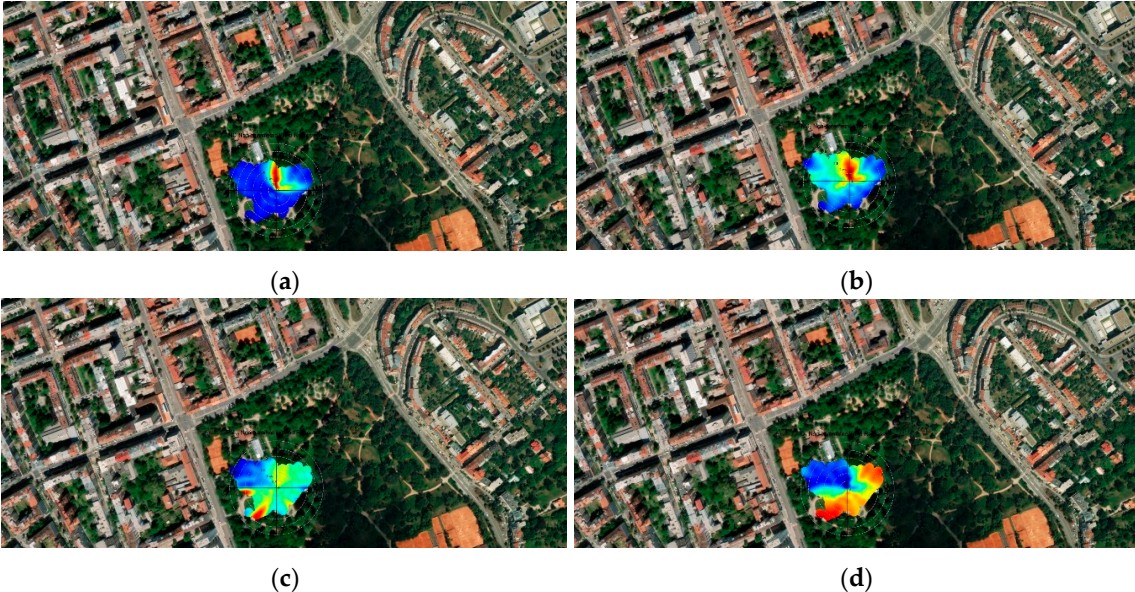

**Figure 14.** The NO (**a**), NO$_2$ (**b**), PM$_{10}$ (**c**), and O$_3$ (**d**) concentration relationships to the wind speed and direction at Lužánky SVC; sampling started on 22.8.2019 7:00.

Figure 14b shows that the highest $NO_2$ concentrations were measured under northern wind directions (Figure 14b). In western to northern wind directions, we established the lowest concentrations of $O_3$ (Figure 14d). The nitrogen oxides had probably originated from the traffic on the road north of the park. The $PM_{10}$ concentrations did not exhibit any significant relationship to the wind direction in this case.

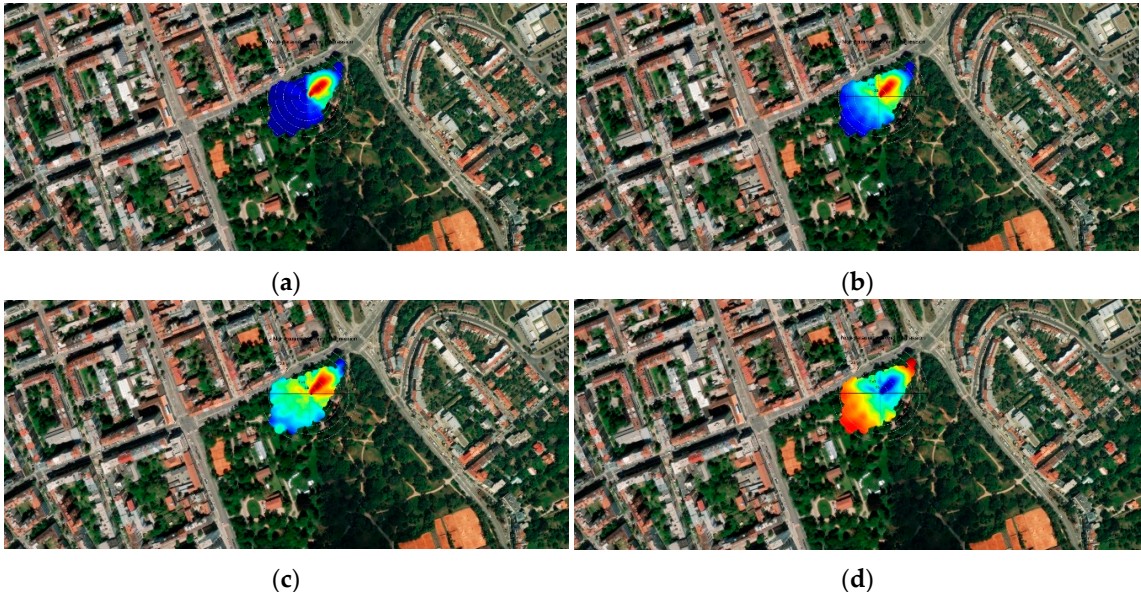

**Figure 15.** The NO (**a**), $NO_2$ (**b**), $PM_{10}$ (**c**), and $O_3$ (**d**) concentration relationships to the wind speed and direction at Lužánky SS; sampling started on 6.3.2019 7:00.

Figure 15a–c indicates that the highest NO, $NO_2$, and $PM_{10}$ concentrations were measured under a northeastern wind direction. In the same wind directions, we acquired the lowest concentrations of $O_3$ (Figure 15d). Both the nitrogen oxides and the $PM_{10}$ had probably been generated by the traffic on the crossroads to the northeast of the park.

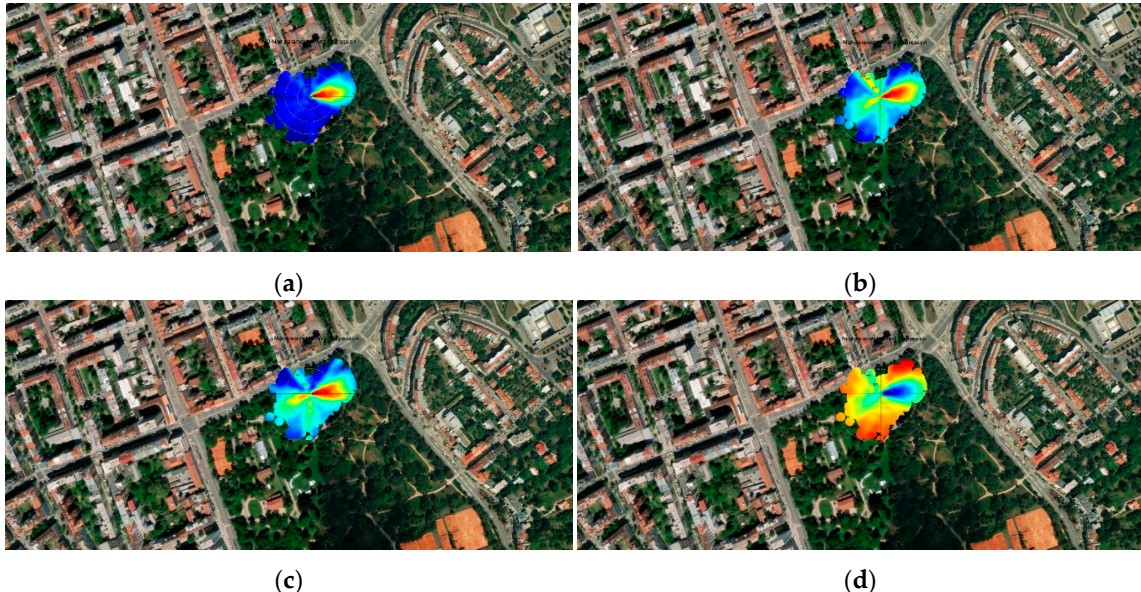

**Figure 16.** The NO (**a**), $NO_2$ (**b**), $PM_{10}$ (**c**), and $O_3$ (**d**) concentration relationships to the wind speed and direction at Lužánky SS; sampling started on 22.8.2019 7:00.

Figure 16 displays a situation similar to that shown in Figure 15. It clearly follows from the images in both of the figures that, at the Lužánky SS locality, the traffic pollution (NO) is contained by the Svojsík srub building. At the Lužánky SVC site (Figures 12–14), conversely, the NO source is blocked by the Leisure Center from the west.

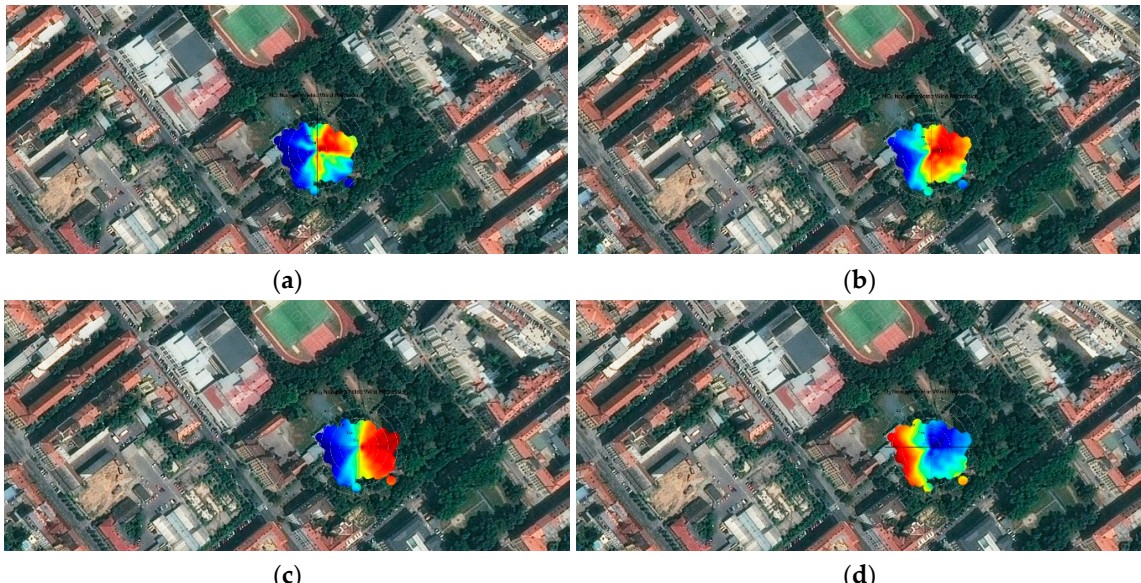

(**a**)　　　　　　　　　　　　　　　　　　　　　　　(**b**)

(**c**)　　　　　　　　　　　　　　　　　　　　　　　(**d**)

**Figure 17.** The NO (**a**), NO$_2$ (**b**), PM$_{10}$ (**c**), and O$_3$ (**d**) concentration relationships to the wind speed and direction at Tyršův sad; sampling started on 8.2.2019 7:00.

There are no significant transport-based air pollution sources near Tyršův sad; the air pollution at this location can be rather generated by long-distance transfer or, especially in wintertime, PM$_{10}$ from local heating. Figure 17 shows the pollution from eastern directions, and Figure 18 displays the ambiguous situation at the site.

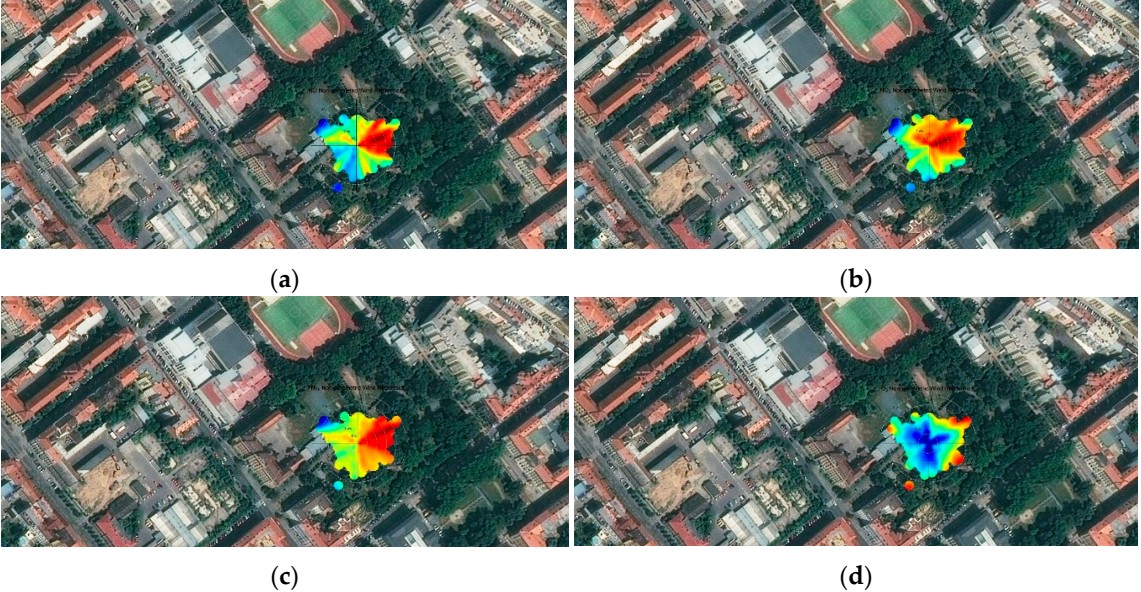

(**a**)　　　　　　　　　　　　　　　　　　　　　　　(**b**)

(**c**)　　　　　　　　　　　　　　　　　　　　　　　(**d**)

**Figure 18.** The NO (**a**), NO$_2$ (**b**), PM$_{10}$ (**c**), and O$_3$ (**d**) concentration relationships to the wind speed and direction at Tyršův sad; sampling started on 2.8.2019 7:00.

As outlined above, the problem of reducing PM concentrations in urban parks has been discussed in diverse papers, e.g., [4–6]. Other articles analyzed the impact of urban greenery on $NO_x$, $NO_2$ [14], and $O_3$ [16]. In this study, the outcomes presented within the referenced research reports are followed and developed through such procedural approaches as monitoring the influence of wind and air temperature on pollutant concentrations. The measurements have shown that, in addition to vegetation, seasonal changes of meteorological conditions and human activities in parks embody a substantial aspect modifying the local situation, as observed at Lužánky park in August 2019. The obtained results have confirmed the conclusions proposed by Kumar et al. [12], namely, that progressive steps need to be taken to bring further knowledge in the field. The relationships between $O_3$, NO, and $NO_2$ were studied by Han et al. [25]; interestingly, the outcomes of our research resemble Han et al.'s findings in suggesting that, as regards the study area(s), the daily NO cycle initiated by flue gas emissions from motor vehicles and continued by the related conversion of the pollutant into $NO_2$, had a major impact on the regular ozone process. The daily course of concentrations in these pollutants was similar, too.

## 4. Conclusions

In four 14-day campaigns, concentrations of NO, $NO_2$, $PM_{10}$, and $O_3$ were measured at five diverse locations, of which four were enclosed within Brno parks and one set at a road adjacent to a park. Compared to the average values, significantly higher nitrogen oxide concentrations were determined at the monitored spots of Koliště and Koliště-road in colder weather. Both of the locations are situated near a busy road exhibiting a traffic intensity of 33,000 vehicles/d. In terms of $PM_{10}$, the highest concentrations were obtained at Koliště park, with an average air temperature that proved to be the lowest among the values adopted for the other measurements. At Lužánky park, the $PM_{10}$ concentrations measured in warmer weather reached higher than those acquired during colder periods—an effect probably caused by the park being a popular public barbecue place. Using the "openairmaps" software package, we determined the directions pointing to the main sources of pollution at the individual spots. Based on this procedure, it was estimated that the main air pollution sources affecting the parks lie in the adjacent roads and crossroads. In some cases, however, human activities of people in the parks (barbecue) can also be regarded as important or semi-critical. By extension, we established that the overall surface layout, prominently including buildings in the park, can locally shield the impact of traffic on the air quality. Interestingly, the air quality in the parks approached that of the urban background locations, except for Koliště park, which, due to its shape and proximity to a very busy road, showed the characteristics of a regular traffic location.

**Author Contributions:** Conceptualization, J.H. (Jiří Huzlík); Data curation, J.H. (Jitka Hegrová) and K.E.; Formal analysis, J.H. (Jiří Huzlík), J.H. (Jitka Hegrová) and K.E.; Funding acquisition, J.H. (Jitka Hegrová) and M.B.; Investigation, J.H. (Jiří Huzlík) and R.L.; Methodology, J.H. (Jiří Huzlík), J.H. (Jitka Hegrová), R.L. and M.B.; Project administration, J.H. (Jitka Hegrová), R.L. and M.B.; Resources, J.H. (Jiří Huzlík), J.H. (Jitka Hegrová) and K.E.; Supervision, J.H. (Jitka Hegrová), R.L. and M.B.; Validation, J.H. (Jiří Huzlík), J.H. (Jitka Hegrová) and R.L.; Visualization, J.H. (Jiří Huzlík) and R.L.; Writing—original draft, J.H. (Jiří Huzlík) and R.L.; Writing—review & editing, J.H. (Jiří Huzlík), J.H. (Jitka Hegrová), K.E. and R.L. All authors have read and agreed to the published version of the manuscript.

**Funding:** This article was produced under support from the Technology Agency of the Czech Republic within the ÉTA framework, project TL01000286, on the research infrastructure acquired from the Operational Programme Research and Development for Innovations (CZ.1.05/2.1.00/03.0064).

**Conflicts of Interest:** The authors declare no conflicts of interest. The funders had no role in the designing of the study; in the collection, analyses, or interpretation of the data; in the writing of the manuscript; or in the decision to publish the results.

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
