# Peer review of "Air Quality in Brno City Parks"

_atmosphere, doi:10.3390/atmos11050510_

Round 1
Reviewer 1 Report
Brief summary
This paper measured the concentrations of nitric oxide (NO), nitrogen dioxide (NO2), ozone (O3) and PM10 in several parks of Brno city in the Czech Republic in different seasons. It aimed to investigate the reduction influence of greenery on air quality in parks and its variation in different seasons. It applied some instruments to measure NO, NO2, O3, and PM10 at the same time, and conducted several campaigns in different seasons. This paper described the distribution patterns of air pollution in these parks and also investigated the impact of wind direction and wind speed on the distributions of air pollutants. However, this paper didn't state the design of the campaigns in detailed and needed to improve the interpretation of the results carefully. It can be considered to publish in the journal of Atmosphere after major revision.
Broad comments
1. Please re-summarize the abstract. Abstract is a summary of the whole paper and should include background, hypothesis, main results and conclusion of the paper. It should not be a display of results and some ambiguous sentences.
2. There should be one paragraph to state the hypothesis or goal of this paper in the part of INTRODUCTION. It is usually in the last paragraph of INTRODUCTION.
3. After stating the purpose of this paper, you also need to describe how you design your campaign to achieve your purpose. For example, why to choose these parks, why to put the equipments in these locations of the parks.
4. It is good that the authors did an external and/or internal calibration for NOx, O3, and PM10. Please show this data and results in the main text or supplementary of this paper. It is very important for proving that the data in this paper is qualified.
5. How or where did you obtain the meteorological data? Please add this part in the main text of this paper.
6. Please describe the characteristics of each monitoring sites (surrounding environments, possible pollution sources, what kind of people would like to come to these parks and the most period in a year, etc.) and how did you choose to place the sites where and why to put the equipments.
7. I would like to suggest the authors to change the second part of the paper to "METHOD". In this part, please describe the equipments, characteristics of monitoring sites, data calibration, the measurement of meteorological data.
8. Please add more interpretation to each figure and table to state what you found from your data. It should not be just one simple sentence.
9. One of the goals of this paper is to investigate the impact of parks on reducing air pollution in cities, so there should be one site close to road to monitor the air quality outside the park and one site in the park to monitor the air quality in the park. And, it should be seen that the air quality in the park is better than the air quality outside the park. You had one roadside site and one inside site in the park of Koliště, but the monitoring durations at the two sites were not simultaneous.
10. The fourth part of this paper-DISCUSSION-is more like a part of results.
Specific comments
- Line 22: the unit of NOx should be μg/m3.
- Please keep the same tense in the whole paper.
- Lines 127-145 should be moved to the part of "METHOD".
- Lines 177-179: why the other two parks didn't have a higher concentration of air pollutants in warmer seasons? People didn't go to these parks for barbecue?
- Lines 185-215 should be moved to the part of "METHOD".
- In Figures 13 to 22, the distribution of O3 is always adverse with NO2. Is it the influence of greenery or the atmospheric chemistry reaction between NO2, O3 and some other pollutants?
- Please find the professional organization or native English friends to help with the writing of this paper.
Author Response
Broad comments
- Please re-summarize the abstract. Abstract is a summary of the whole paper and should include background, hypothesis, main results and conclusion of the paper. It should not be a display of results and some ambiguous sentences.
Abstract was re-summarized
- There should be one paragraph to state the hypothesis or goal of this paper in the part of INTRODUCTION. It is usually in the last paragraph of INTRODUCTION.
A paragraph describing the goal of the paper has been added
- After stating the purpose of this paper, you also need to describe how you design your campaign to achieve your purpose. For example, why to choose these parks, why to put the equipments in these locations of the parks.
A paragraph describing the nature of the parks, their usual use and their location in the city has been added to the Method chapter.
- It is good that the authors did an external and/or internal calibration for NOx, O3, and PM10. Please show this data and results in the main text or supplementary of this paper. It is very important for proving that the data in this paper is qualified.
PM calibration data was added as part of the supplementary of this paper. Calibration when measuring nitrogen oxide concentrations consists in setting the zero and span on the calibration gas, internal calibration for ozone consists in setting the zero and span using a built-in ozone generator. Regular annual calibrations of nitrogen and ozone oxide analyzers are performed in an external accredited calibration laboratory. Regular calibrations of meteorological parameters are also performed every two years in an external accredited calibration laboratory.
- How or where did you obtain the meteorological data? Please add this part in the main text of this paper.
A paragraph describing meteorological data obtaining has been added to the Method chapter
- Please describe the characteristics of each monitoring sites (surrounding environments, possible pollution sources, what kind of people would like to come to these parks and the most period in a year, etc.) and how did you choose to place the sites where and why to put the equipments.
A description of site characteristics has been added to the Method chapter
- I would like to suggest the authors to change the second part of the paper to "METHOD". In this part, please describe the equipments, characteristics of monitoring sites, data calibration, the measurement of meteorological data.
The second part of the paper was changed to "METHOD". Descriptions of localities, used methods, calibrations, measurements of meteorological data were placed here.
- Please add more interpretation to each figure and table to state what you found from your data. It should not be just one simple sentence.
Figures and tables were rearranged, interpretations were extended
- One of the goals of this paper is to investigate the impact of parks on reducing air pollution in cities, so there should be one site close to road to monitor the air quality outside the park and one site in the park to monitor the air quality in the park. And, it should be seen that the air quality in the park is better than the air quality outside the park. You had one roadside site and one inside site in the park of Koliště, but the monitoring durations at the two sites were not simultaneous.
Data from measurements at reference stations - automatic air pollution monitoring stations, characterizing background and traffic locations - were included.
- The fourth part of this paper-DISCUSSION-is more like a part of results.
Discussion and Results were merged.
Specific comments
- Line 22: the unit of NOx should be μg/m3.
Fixed
- Please keep the same tense in the whole paper.
Fixed
- Lines 127-145 should be moved to the part of "METHOD".
Rearranged
- Lines 177-179: why the other two parks didn't have a higher concentration of air pollutants in warmer seasons? People didn't go to these parks for barbecue?
This was explained in the text (people went only to Lužánky for barbecue)
- Lines 185-215 should be moved to the part of "METHOD".
Fixed
- In Figures 13 to 22, the distribution of O3 is always adverse with NO2. Is it the influence of greenery or the atmospheric chemistry reaction between NO2, O3 and some other pollutants?
Explained in text of paper. It was influence of atmospheric chemistry reaction
- Please find the professional organization or native English friends to help with the writing of this paper.
It will be done in the final revision
Reviewer 2 Report
Thank you for the opportunity to review this manuscript.
Review of Huzlik et al. Air quality in Brno city parks
I have read with great interest the paper by Huzlik et al. I recommend publication after some revision. I have outlined a number of points below.
- The English is good, but really needs to be read carefully by an English editor. There are numerous grammar issues, nothing serious, but it is in need of some editing.
- The Introduction is very long and presents all of the background information. However, the Introduction does a poor job of introducing the authors interesting work as the authors’ study is mixed in with the background. I suggest shortening the Introduction to something like as follows at the end of this review (I rearranged some of the authors’ text below),
- I suggest adding a section on “Background” following the Introduction. If the journal would prefer the Introduction and Background to be in one section, then the authors study needs to be reorganized into one or two consecutive paragraphs.
- Some references are cited inconsistently, it would be good to be consistent.
- The Background omits a great deal of the literature on the uptake of gases by vegetation. There have been decades of work on uptake of acidifying gases by vegetation. For example, Baldocchi, 1988; Schaefer et al., 1992a, 1992b; Draaijers et al., 1997; Neal, 2002; Wesely, 1989; Yang et al., 2005; Nowak et al., 1998, 2006; Mooers and Massman, 2017. It is not necessary that the authors go into detail about uptake of acidifying gases, however, they should be aware of the body of literature.
- The Experiments section (lines 103-125) begins to lay out the methods, however, much of the actual methods that were performed are found in the Results section.
- I would suggest moving all of the methods up to the Experiments section and rename that section “Methods.”
- The location map should be referenced in this section. I cannot tell from the Map which locations are which. Can they be labeled better?
- Then include all of the information on measurement of PM and NOx, O3, wind speed, etc. in the Methods section.
- Discuss the statistical techniques that will be used
- The first part of the Results section from lines 127-145 is really methods.
- In the Results section, it might be helpful to combine figures 5 - 10 from the results section on a single page.
Sample Introduction (from the authors work but reorganized)
Urban green spaces, city parks, are very often considered as localities with the best air quality in a city and thus are often used by citizens for relaxation and active recreation. However, there are very few studies supporting this generally accepted claim.
Air pollution and human health, as well as green infrastructure and human health, are often studied together. Linking green infrastructure with air quality and human health together is aspect of review of Kumar et al. [12]. They conclude that urban vegetation can have health benefits, but there is a little knowledge about air pollution reduction by urban vegetation and its benefits and it is necessary to do steps to gain further knowledge. Air pollution presents a major risk to human health, causes premature deaths and can reduce quality of life. Quantifying the role of vegetation in reducing air pollution concentrations is important. Most current methods to calculate pollution removal are static and do not represent atmospheric transport of pollutants, or pollutants and meteorology interaction.
Because of the gap of information on air quality in city parks, our study presents results of measurements of gaseous pollutants concentration and PM10 concentrations in the Brno city that is the second largest city in the Czech Republic with a population of approximately 377,000. These results are compared with measurements in other city parts and possible pollutant sources in parks are discussed. PM10 solids concentrations were determined continuously by nephelometric method followed by gravimetric method validation. We evaluate air quality within the local environment by correlation with measurements of wind direction, wind speed, temperature and relative humidity to identify potential sources of air pollution in parks. The “openair” and “openairmaps” packages from the OpenSource software R 14 were used to analyze the effects of meteorological conditions on air pollution. Local polar concentration maps are used to localize of the most serious sources of air pollution in urban parks. The results of the analyzes show that the majority of polluters at the measuring point are most likely crossroads near the sampled localities.

Author Response
1. The English is good, but really needs to be read carefully by an English editor. There are numerous grammar issues, nothing serious, but it is in need of some editing.
It will be done in the final revision
2. The Introduction is very long and presents all of the background information. However, the Introduction does a poor job of introducing the authors interesting work as the authors’ study is mixed in with the background. I suggest shortening the Introduction to something like as follows at the end of this review (I rearranged some of the authors’ text below),
Introduction has been rearranged, but not shortened (compromise with Reviewer 1)
3. I suggest adding a section on “Background” following the Introduction. If the journal would prefer the Introduction and Background to be in one section, then the authors study needs to be reorganized into one or two consecutive paragraphs.
Introduction has been rearranged, paragraphs with background were added
4. Some references are cited inconsistently, it would be good to be consistent.
1. The Background omits a great deal of the literature on the uptake of gases by vegetation. There have been decades of work on uptake of acidifying gases by vegetation. For example, Baldocchi, 1988; Schaefer et al., 1992a, 1992b; Draaijers et al., 1997; Neal, 2002; Wesely, 1989; Yang et al., 2005; Nowak et al., 1998, 2006; Mooers and Massman, 2017. It is not necessary that the authors go into detail about uptake of acidifying gases, however, they should be aware of the body of literature.
Thank you for providing data on literary sources, we will use them to continue solving our project.
5. The Experiments section (lines 103-125) begins to lay out the methods, however, much of the actual methods that were performed are found in the Results section.
1. I would suggest moving all of the methods up to the Experiments section and rename that section “Methods.”
The paper was rearranged according to the comments
2. The location map should be referenced in this section. I cannot tell from the Map which locations are which. Can they be labeled better?
Figure 4 was rearranged
3. Then include all of the information on measurement of PM and NOx, O3, wind speed, etc. in the Methods section.
The paper was rearranged according to the comments
4. Discuss the statistical techniques that will be used
The statistical methods used were described in the METHOD chapter
5. The first part of the Results section from lines 127-145 is really methods.
The paper was rearranged according to the comments
6. In the Results section, it might be helpful to combine figures 5 - 10 from the results section on a single page.
The paper was rearranged according to the comments
Round 2
Reviewer 1 Report
The authors made a big progress on this paper. It can be considered to publish in the journal of Atmosphere.
Author Response
The English was revised by a professional agency